# Computationally Efficient Direction Finding for Conformal MIMO Radar

**DOI:** 10.3390/s24186065

**Published:** 2024-09-19

**Authors:** Haochen Wang, Zhiyu Yu, Fangqing Wen

**Affiliations:** 1Department of Communication Technologies and System Design, Technical University of Denmark (DTU), 2800 Lyngby, Denmark; s232176@dtu.dk; 2School of Microelectronics and Communication Engineering, Chongqing University, Chongqing 401331, China; 30025039@alu.cqu.edu.cn; 3College of Computer and Information Technology, China Three Gorges University, Yichang 443002, China

**Keywords:** conformal array, MIMO radar, DOD, DOA, rotationally invariant technique

## Abstract

The use of conformal arrays offers a significant advancement in Multiple-Input–Multiple-Output (MIMO) radar, enabling the placement of antennas on irregular surfaces. For joint Direction-of-Departure (DOD) and Direction-of-Arrival (DOA) estimation in conformal-array MIMO radar, the current spectrum-searching methods are computationally too expensive, while the existing rotation-invariant method may suffer from phase ambiguity caused by the non-Nyquist spacing of the sensors. In this paper, an improved rotationally invariant technique is proposed. The core function of the proposed algorithm is to estimate the phase differences between the adjacent sensors; then, it eliminates phase ambiguity via the previous estimated standard phase difference. Thereafter, DODs and DOAs are obtained via Least Squares (LS) fitting. The proposed method provides closed-form estimates for joint DOD and DOA estimation, which is much more efficient than the existing spectrum-searching techniques. Numerical simulations show that the proposed algorithm can accurately determine 2D DODs and DOAs of targets, only requiring approximately 1% of the running time required by existing spectrum-searching approaches.

## 1. Introduction

Multiple-Input–Multiple-Output (MIMO) radar has been well recognized as a breakthrough technique in next-generation radar and communication systems [1,2,3,4,5,6,7,8,9,10]. Unlike traditional phased-array radar, MIMO radar transmits mutual orthogonal waveforms via multiple active antennas, enabling the radar system to exhibit both spatial and temporal diversities. As a result, MIMO radar offers more Degrees of Freedom (DOFs) than phased-array radar and yields significant improvements in spatial resolution, target detection capability, and overall system performance [11].

The estimation of the direction of departure (DOD) and direction of arrival (DOA) in MIMO radar is a critical task that is pivotal for accurately determining the position and movement of targets. To date, a variety of estimators has been developed [12,13], such as the Estimation of Signal Parameters via Rotational Invariance Technique (ESPRIT) [14,15], Multiple Signal Classification (MUSIC) [16,17], the maximum likelihood method [18], the matrix pencil method [19], the rooting method [20], and the tensor algorithm [21,22,23,24]. Existing estimators can be divided into two categories according to whether they can provide closed-form estimates. The first category includes the spectrum-searching strategies, e.g., Reduced-Dimension MUSIC (RD-MUSIC) [16] and maximum likelihood [18], which is suitable for an arbitrary array geometry. Usually, a cost function such as the Frobenius norm or logarithm is established, and the spectrum-searching strategy obtains estimates by finding the local minima/maxima of the cost function. Owing to the exhaustive nature of the search calculation, spectrum-searching strategies are often computationally inefficient and may suffer from off-grid issues, as the search grid may not match the true values. The second category comprises closed-form solutions, typical representatives of which are ESPRIT, least squares (LS) fitting, and the rooting method. Although closed-form solutions are often much more efficient than peak searching methods, they require linear or planar array geometries [25,26]. Among the various closed-form estimators, ESPRIT is one of the most attractive. Usually, ESPRIT offers less accurate results than MUSIC, since it may sacrifice the effective array aperture. Nevertheless, it is much more efficient than MUSIC, since the former does not require a searching calculation.

Conformal-array MIMO radar represents a significant advancement in radar technology, enabling the placement of antennas on curved surfaces, such as the fuselage of an aircraft or the hull of a ship. This flexibility allows for improved coverage and a reduced radar cross-section, thereby enhancing the stealth and efficiency of the radar system. The unique geometrical configuration of conformal arrays, however, introduces new challenges in signal processing, particularly in the joint estimation of DOD and DOA. To the best of our knowledge, only a few works have been focused on signal processing for conformal-array MIMO radar [27,28,29]. For MIMO radar with conformal Tx/Rx arrays, existing spectrum-searching methods such as MUSIC can be directly applied. Like the previously mentioned counterparts, however, spectrum-searching methods cannot avoid a heavy computational burden, especially in high-dimensional direction finding (i.e., two-dimensional angle estimation). Although reduced dimension (RD) techniques, e.g., RD-MUSIC, can relax the computational burden [16], they may fail to work in the LS procedure due to the inter-element space being too large. Another alternative is the interpolation-based approach [30], which can construct a virtual uniform-array geometry. However, the interpolation-based approach requires prior knowledge of the incident target spatial sector, which is usually unavailable to radar systems. Besides, the calculation of the interpolation function maybe computationally inefficient for real-time applications. A polarized array [31,32] is another possible candidate to avoid the abovementioned drawbacks, but the vector components may suffer from serious mutual coupling, offering inaccurate estimation results. To the best of our knowledge, the use of closed-form 2D direction finding for conformal scalar-array MIMO has only been discussed in [33]. Therein, the authors proposed an ESPRIT estimator for an arbitrary array geometry; however, the proposed estimator requires the inter-element space to be no larger than a half wavelength. Closed-form methods may not be directly applicable to conformal arrays due to their non-uniform and potentially complex shapes. Consequently, there is a need for tailored algorithms that can effectively handle the intricacies of conformal-array geometries while exploiting the spatial diversity and waveform diversity inherent in MIMO radar systems.

Although the rotationally invariant property does not exist for a conformal array, it may hold for a visual array in conformal-array MIMO radar. Such a characteristic can help us to find a quick solution for DOD/DOA estimation. This paper revisits the problem of joint DOD and DOA estimation in conformal-array MIMO radar systems. An improved ESPRIT framework is introduced to provide a closed-form solution. The kernel idea of the proposed framework is to estimate the phase differences between adjacent sensors. To avoid the phase ambiguity caused by large inter-element spacing, a phase compensation strategy is proposed. Phase ambiguity is eliminated by using the estimated phase differences associated with sensors no farther apart than a half wavelength. The proposed algorithm leverages the unique characteristics of conformal arrays and the advantages of MIMO radar to achieve high-resolution and accurate estimation of target angles. It ensures robust performance, even in challenging scenarios with multiple targets. Herein, detailed analyses are presented and numerical results are designed to show the advantages of the proposed algorithm.

Notations. Capital letters, e.g., X, and lowercase letters, e.g., x, in bold denote matrices and vectors, respectively. Superscript ${\left(\cdot \right)}^{T}$, ${\left(\cdot \right)}^{H}$, ${\left(\cdot \right)}^{-1}$, and ${\left(\cdot \right)}^{†}$ represent the operations of transpose, Hermitian transpose, inverse, and pseudo-inverse, respectively. ⊗ and ⊕ stand for the Kronecker product and Hadamard product, respectively. $I_{N}$ represents an N×N identical matrix. ${\left\|X\right\|}_{*}$ and ${\left\|X\right\|}_{F}$ denote the nuclear norm (sum of the singular values) and Forbenius norm of X, respectively. $\mathrm{diag}\left\{r\right\}$ and vec(r) denote the diagonalization operation and the vectorization operation, respectively. $E\left\{\cdot \right\}$ and $\mathrm{rank}\left\{\cdot \right\}$ return the expectation of a variable and the rank of a matrix, respectively. $\delta \left(\cdot \right)$ is the Kronecker delta, and $\mathrm{round}\left\{\cdot \right\}$ returns the nearest integer.

## 2. Signal Model

Consider a bistatic MIMO radar system with conformal arrays at both the transmitting (Tx) end and the receiving (Rx) end. Suppose that there is an *M*-element Tx array and an *N*-element Rx array, as shown in Figure 1. Assume that there are *K* point-like, non-coherent targets appearing in the far field of the MIMO radar. $\left(\theta_{t,k},\phi_{t,k}\right)$ denotes the 2D DOD corresponding to the *k*-th target, and $\left(\theta_{r,k},\phi_{r,k}\right)$ denotes the 2D DOA corresponding to the *k*-th target, where $\theta_{t,k}$ and $\theta_{r,k}$ are the elevation angles and $\phi_{t,k}$ and $\phi_{r,k}$ are the azimuth angles. Also assume that the carrier wavelength is λ. Set the first Tx/Rx sensor as the reference located at the origin. As a result, the respecitve time-of-arrival differences with respect to the *m*-th Tx sensor and the *n*-th Rx sensors are
(1a)$$\begin{array}{ccc} d_{t,m,k} & = & x_{t,m}\cos\left(\phi_{t,k}\right)\sin\left(\theta_{t,k}\right)+y_{t,m}\sin\left(\phi_{t,k}\right)\sin\left(\theta_{t,k}\right)+z_{t,m}\cos\left(\theta_{t,k}\right)\\ & = & p_{t,m}^{T}c_{t,k} \end{array}$$
(1b)$$\begin{array}{ccc} d_{r,n,k} & = & x_{r,n}\cos\left(\phi_{r,k}\right)\sin\left(\theta_{r,k}\right)+y_{r,n}\sin\left(\phi_{r,k}\right)\sin\left(\theta_{r,k}\right)+z_{r,n}\cos\left(\theta_{r,k}\right)\\ & = & p_{r,n}^{T}c_{r,k}, \end{array}$$
where $p_{t,m}$ and $p_{r,n}$ denote the coordinate vectors of the *m*-th Tx sensor and the *n*-th Rx sensor, respectively. $c_{t,k}$ and $c_{r,k}$ are direction cosines corresponding to the Tx array and the Rx array, respectively. In detail, $p_{t,m}$, $p_{r,n}$, $c_{t,k}$, and $c_{r,k}$ are expressed as
(2a)$$p_{t,m}={\left[x_{t,m},y_{t,m},z_{t,m}\right]}^{T}$$
(2b)$$p_{r,n}={\left[x_{r,n},y_{r,n},z_{r,n}\right]}^{T}$$
(2c)$$c_{t,k}={\left[\cos\left(\phi_{t,k}\right)\sin\left(\theta_{t,k}\right),\sin\left(\phi_{t,k}\right)\sin\left(\theta_{t,k}\right),\cos\left(\theta_{t,k}\right)\right]}^{T}$$
(2d)$$c_{r,k}={\left[\cos\left(\phi_{r,k}\right)\sin\left(\theta_{r,k}\right),\sin\left(\phi_{r,k}\right)\sin\left(\theta_{r,k}\right),\cos\left(\theta_{r,k}\right)\right]}^{T},$$
respectively.

Accompanied by the narrow-band assumption, the associated Tx/Rx spatial response vectors are expressed as
(3a)$$a_{t}\left(\theta_{t,k},\phi_{t,k}\right)={\left[1,e^{j2\pi d_{t,2,k}/\lambda },\cdots ,e^{j2\pi d_{t,M,k}/\lambda }\right]}^{T}$$
(3b)$$a_{r}\left(\theta_{r,k},\phi_{r,k}\right)={\left[1,e^{j2\pi d_{r,2,k}/\lambda },\cdots ,e^{j2\pi d_{r,N,k}/\lambda }\right]}^{T}$$

In addition, we assume the Tx sensors emit mutual orthogonal waveforms (${\left\{w_{m}\left(t\right)\right\}}_{m=1}^{M}$), i.e., $\int_{T_{p}}w_{m}\left(t\right)w_{n}^{*}\left(t\right)dt=\delta \left(m-n\right)$, where *t* is the fast time index (time index during a radar pulse) and $T_{p}$ is the pulse duration. Moreover, the target reflection coefficients (${\left\{\alpha_{k}\left(\tau \right)\right\}}_{k=1}^{K}$) should be at a constant pulse duration, where τ denotes the pulse index. Thus the reflected signal from the *k*-th target can be expressed as [11]
(4)$$r_{k}\left(t,\tau \right)=\alpha_{k}\left(\tau \right)a_{t}^{T}\left(\theta_{t,k},\phi_{t,k}\right)w\left(t\right)$$
where $w\left(t\right)={\left[w_{1}\left(t\right),w_{2}\left(t\right),\cdots ,w_{M}\left(t\right)\right]}^{T}$ denotes the Tx waveform vector. The noisy signal at the Rx end is expressed as [11]
(5)$$x\left(t,\tau \right)=\sum_{k=1}^{K}\alpha_{k}\left(\tau \right)a_{r}\left(\theta_{r,k},\phi_{r,k}\right)a_{t}^{T}\left(\theta_{t,k},\phi_{t,k}\right)w\left(t\right)+e\left(t,\tau \right)$$
where $e\left(t,\tau \right)$ denotes the additive noise vector, which is assumed to satisfy the white Gaussian distribution with a variance of $\sigma^{2}$, namely,
(6)$$E\left\{e\left(t,\tau \right)e^{H}\left(t,\tau \right)\right\}=\sigma^{2}I_{N}$$

After matched filtering ($x\left(t,\tau \right)$) using $w_{m}\left(t\right)$, one obtains
(7)$$\begin{array}{ccc} y_{m}\left(\tau \right) & = & \int_{T_{p}}x\left(t,\tau \right)w_{m}^{*}\left(t\right)dt\\ & = & \sum_{k=1}^{K}\alpha_{k}\left(\tau \right)a_{r}\left(\theta_{r,k},\phi_{r,k}\right)a_{t}^{T}\left(\theta_{t,k},\phi_{t,k}\right)\int_{T_{p}}w\left(t\right)w_{m}^{*}\left(t\right)dt+\int_{T_{p}}e\left(t,\tau \right)w_{m}^{*}\left(t\right)dt\\ & = & \sum_{k=1}^{K}a_{t,m}\left(\theta_{t,k},\phi_{t,k}\right)\alpha_{k}\left(\tau \right)a_{r}\left(\theta_{r,k},\phi_{r,k}\right)+e_{m}\left(\tau \right) \end{array}$$
where $a_{t,m}\left(\theta_{t,k},\phi_{t,k}\right)$ denotes the *m*-th entity of $a_{t}\left(\theta_{t,k},\phi_{t,k}\right)$, $e_{m}\left(\tau \right)=\int_{T_{p}}e\left(t,\tau \right)w_{m}^{*}\left(t\right)dt$. Stacking ${\left\{y_{m}\left(\tau \right)\right\}}_{m=1}^{M}$ into a tall vector yields
(8)$$\begin{array}{ccc} y\left(\tau \right) & = & \sum_{k=1}^{K}\left[a_{t}\left(\theta_{t,k},\phi_{t,k}\right)⊗a_{r}\left(\theta_{r,k},\phi_{r,k}\right)\right]\alpha_{k}\left(\tau \right)+n\left(\tau \right)\\ & = & \left[A_{t}⊙A_{r}\right]s\left(\tau \right)+n\left(\tau \right)\\ & = & As\left(\tau \right)+n\left(\tau \right) \end{array}$$
where $n\left(\tau \right)={\left[n_{1}^{T}\left(\tau \right),n_{2}^{T}\left(\tau \right),\cdots ,n_{M}^{T}\left(\tau \right)\right]}^{T}$ denotes the noise vector after matched filtering, $A_{t}=\left[a_{t}\left(\theta_{t,1},\phi_{t,1}\right),a_{t}\left(\theta_{t,2},\phi_{t,2}\right),\cdots ,a_{t}\left(\theta_{t,K},\phi_{t,K}\right)\right]$ denotes the Tx response matrix, $A_{r}=\left[a_{r}\left(\theta_{r,1},\phi_{r,1}\right),a_{r}\left(\theta_{r,2},\phi_{r,2}\right),\cdots ,a_{r}\left(\theta_{r,K},\phi_{r,K}\right)\right]$ denotes the Rx response matrix, and $A=A_{t}⊙A_{r}$ denotes the virtual response matrix, $s\left(\tau \right)={\left[\alpha_{1}\left(\tau \right),\alpha_{2}\left(\tau \right),\cdots ,\alpha_{k}\left(\tau \right)\right]}^{T}$. Suppose that the targets are uncorrelated; then, its covariance matrix is a diagonal matrix, i.e.,
(9)$$\begin{array}{ccc} R_{s} & = & E\left\{s\left(\tau \right)s^{H}\left(\tau \right)\right\}\\ & = & \mathrm{diag}\left\{\beta_{1},\beta_{2},\cdots ,\beta_{K}\right\} \end{array}$$
where $\beta_{k}$ represents the power of $\alpha \left(\tau \right)$. On the other hand, one can observe when that $n\left(\tau \right)=\int_{T_{p}}w^{*}\left(t\right)⊗e\left(t,\tau \right)dt$, the covariance matrix of $n\left(\tau \right)$ is
(10)$$\begin{array}{ccc} R_{n} & = & E\left\{n\left(\tau \right)n^{H}\left(\tau \right)\right\}\\ & = & E\left\{\int_{T_{p}}\int_{T_{p}}\left[w^{*}\left(t_{1}\right)⊗e\left(t_{1},\tau \right)\right]\left[w^{T}\left(t_{2}\right)⊗e^{H}\left(t_{2},\tau \right)\right]dt_{1}dt_{2}\right\}\\ & = & \int_{T_{p}}\int_{T_{p}}{\left[w\left(t_{1}\right)w^{H}\left(t_{2}\right)\right]}^{*}⊗E\left\{e\left(t_{1},\tau \right)e\left(t_{2},\tau \right)\right\}dt_{1}dt_{2}\\ & = & \sigma^{2}I_{MN} \end{array}$$

Consequently, the covariance matrix of $y\left(\tau \right)$ is
(11)$$\begin{array}{ccc} R_{y} & = & E\left\{y\left(\tau \right)y^{H}\left(\tau \right)\right\}\\ & = & AR_{s}A^{H}+\sigma^{2}I_{MN} \end{array}$$

In practice, $R_{y}$ can be estimated via actual collected measurements as
(12)$${\hat{R}}_{y}=\frac{1}{P}\sum_{\tau =1}^{P}y\left(\tau \right)y^{H}\left(\tau \right)$$
where *P* denotes the number of measurements. Obviously, by performing eigendecomposition of $R_{y}$, one can obtain the signal subspace ($E_{s}$), which consists of the eigenvectors corresponding to the *K* dominate eigenvalues. It is well known to us that $E_{s}$ spans the same subspace as A, i.e.,
(13)$$E_{s}=\mathrm{AT}$$
where T is a K×K non-singular matrix.

Most existing subspace-based estimators, e.g., MUSIC and ESPRIT, rely on the prior determination of *K*. In this paper, we assume that *K* is known to us; otherwise, the algorithm is invalid. The accurate determination of *K* is an opening topic but beyond the scope of this paper.

## 3. The Proposed Method

### 3.1. Overview of ESPRIT

The existing ESPRIT method is based on uniformity assumptions with respect to both the Tx array and the Rx array. In what follows, we provide a brief overview of ESPRIT. We define the following selective matrices:
(14a)$$J_{\tilde{M}1}=\left[I_{\tilde{M}-1},0_{\tilde{M}-1}\right]$$
(14b)$$J_{\tilde{M}2}=\left[0_{\tilde{M}-1},I_{\tilde{M}-1}\right]$$

For a $\tilde{M}\times \tilde{N}$ URA geometry of the Tx array and an inter-element spacing of *d* (d≤λ/2), the Tx response matrix can be expressed as
(15)$$a_{t}\left(\theta_{t,k},\phi_{t,k}\right)=a_{t,x}\left(\theta_{t,k},\phi_{t,k}\right)⊗a_{t,y}\left(\theta_{t,k},\phi_{t,k}\right)$$
with
(16a)$$a_{t,x}\left(\theta_{t,k},\phi_{t,k}\right)={\left[1,e^{j2\pi d\cos\left(\phi_{t,k}\right)\sin\left(\theta_{t,k}\right)/\lambda },\cdots ,e^{j2\pi \left(\tilde{M}-1\right)\cos\left(\phi_{t,k}\right)\sin\left(\theta_{t,k}\right)d/\lambda }\right]}^{T}$$
(16b)$$a_{t,y}\left(\theta_{t,k},\phi_{t,k}\right)={\left[1,e^{j2\pi d\sin\left(\phi_{t,k}\right)\sin\left(\theta_{t,k}\right)/\lambda },\cdots ,e^{j2\pi \left(\tilde{N}-1\right)\sin\left(\phi_{t,k}\right)\sin\left(\theta_{t,k}\right)d/\lambda }\right]}^{T}$$

Therefore, the uniformity of the Tx array can be expressed as
(17a)$$\left[J_{\tilde{M}1}⊗I_{\tilde{N}}\right]A_{t}\Phi_{t,u}=\left[J_{\tilde{M}2}⊗I_{\tilde{N}}\right]A_{t}$$
(17b)$$\left[I_{\tilde{M}}⊗J_{\tilde{N}1}\right]A_{t}\Phi_{t,v}=\left[I_{\tilde{M}}⊗J_{\tilde{N}2}\right]A_{t}$$
with
(18a)$$\Phi_{t,u}=\mathrm{diag}\left\{e^{-j2\pi du_{t,1}/\lambda },e^{-j2\pi du_{t,2}/\lambda },\cdots ,e^{-j2\pi du_{t,K}/\lambda }\right\}$$
(18b)$$\Phi_{t,v}=\mathrm{diag}\left\{e^{-j2\pi dv_{t,1}/\lambda },e^{-j2\pi dv_{t,1}/\lambda },\cdots ,e^{-j2\pi dv_{t,K}/\lambda }\right\}$$
where $u_{t,k}=\cos\left(\phi_{t,k}\right)\sin\left(\theta_{t,k}\right)$ and $v_{t,k}=\sin\left(\phi_{t,k}\right)\sin\left(\theta_{t,k}\right)$, respectively. This should yield
(19a)$$\left[J_{\tilde{M}1}⊗I_{\tilde{N}}⊗I_{N}\right]A\Phi_{t,u}=\left[J_{\tilde{M}2}⊗I_{\tilde{N}}⊗I_{N}\right]A$$
(19b)$$\left[I_{\tilde{M}}⊗J_{\tilde{N}1}⊗I_{N}\right]A\Phi_{t,v}=\left[I_{\tilde{M}}⊗J_{\tilde{N}2}⊗I_{N}\right]A$$

Replacing A with $E_{s}T^{-1}$, (19) can be rewritten as
(20a)$$\left[J_{\tilde{M}1}⊗I_{\tilde{N}}⊗I_{N}\right]E_{s}T^{-1}\Phi_{t,u}T=\left[J_{\tilde{M}2}⊗I_{\tilde{N}}⊗I_{N}\right]E_{s}$$
(20b)$$\left[I_{\tilde{M}}⊗J_{\tilde{N}1}⊗I_{N}\right]E_{s}T^{-1}\Phi_{t,v}T=\left[I_{\tilde{M}}⊗J_{\tilde{N}2}⊗I_{N}\right]E_{s}$$

Equivalently,
(21a)$$T^{-1}\Phi_{t,u}T={\left(\left[J_{\tilde{M}1}⊗I_{\tilde{N}}⊗I_{N}\right]E_{s}\right)}^{†}\left[J_{\tilde{M}2}⊗I_{\tilde{N}}⊗I_{N}\right]E_{s}$$
(21b)$$\Phi_{t,v}=T{\left(\left[J_{\tilde{M}1}⊗I_{\tilde{N}}⊗I_{N}\right]E_{s}\right)}^{†}\left[J_{\tilde{M}2}⊗I_{\tilde{N}}⊗I_{N}\right]E_{s}T^{-1}$$

Obviously, the diagonal entities of $\Phi_{t,u}$ are the eigenvalues of ${\left(\left[J_{\tilde{M}1}⊗I_{\tilde{N}}⊗I_{N}\right]E_{s}\right)}^{†}\left[J_{\tilde{M}2}\right.$$\left.⊗I_{\tilde{N}}⊗I_{N}\right]E_{s}$, and the *k*-th column vector of T is an eigenvector corresponding to the *k*-th eigenvalue of the latter. By perform eigendecomposition on ${\left(\left[J_{\tilde{M}1}⊗I_{\tilde{N}}⊗I_{N}\right]E_{s}\right)}^{†}$$\left[J_{\tilde{M}2}⊗I_{\tilde{N}}⊗I_{N}\right]E_{s}$, we can obtain the eigenvectors ($\hat{T}$, i.e., the estimated T) and the estimated $\Phi_{t,u}$, as denoted by ${\hat{\Phi }}_{t,u}=\mathrm{diag}\left\{{\hat{u}}_{t,1},{\hat{u}}_{t,2},\cdots ,{\hat{u}}_{t,K}\right\}$. Since $\Phi_{t,u}$ and $\Phi_{t,v}$ are the eigenvalues of $\Psi_{t}$, the diagonal elements of ${\hat{\Theta }}_{t}$ are the same as the diagonal entities in $\Theta_{t}$. In a similar way, we can obtain an estimate of $\Phi_{t,v}$. However, as matrix decompositions are not always unique, the positions of the estimated diagonal entities of $\Phi_{t,u}$ and $\Phi_{t,v}$ may be different. Since we previously estimated T, we can directly calculate the right side of ([Disp-formula FD21b-sensors-24-06065]) to obtain an estimate of $\Phi_{t,v}$ using the estimated T. The estimate is denoted as ${\hat{\Phi }}_{t,v}=\mathrm{diag}\left\{{\hat{v}}_{t,1},{\hat{v}}_{t,2},\cdots ,{\hat{v}}_{t,K}\right\}$. As a result, the diagonal entities (${\hat{u}}_{t,k}$ and ${\hat{v}}_{t,k}$) are a one-to-one mapping. Finally, the 2D DOD can be estimated via
(22a)$${\hat{\phi }}_{t,k}=\arctan\left\{{\hat{v}}_{t,k}/{\hat{u}}_{t,k}\right\}$$
(22b)$${\hat{\theta }}_{t,k}=\arcsin\sqrt{{\hat{u}}_{t,k}^{2}+{\hat{v}}_{t,k}^{2}}$$

Since ${\hat{u}}_{t,k}$ and ${\hat{v}}_{t,k}$ are a one-to-one mapping, ${\hat{\phi }}_{t,k}$ and ${\hat{\theta }}_{t,k}$ are paired automatically. Similarly, we can obtain the 2D DOA in the presence of a URA array geometry.

However, the algorithmic steps would fail in the presence of an arbitrary array geometry. In what follows, we focus on how to obtain closed-form solutions for such a situation.

### 3.2. The Proposed Normalized ESPRIT Algorithm

First, we define the following selective matrices:
(23a)$$J_{t,m}=i_{M,m}⊗I_{N}$$
(23b)$$J_{r,n}=I_{M}⊗i_{N,n}$$

Obviously, we have
(24a)$$J_{t,m}A=A_{t,m}⊙A_{r}$$
(24b)$$J_{r,n}A=A_{t}⊙A_{r,n}$$
where $A_{t,m}$ and $A_{r,n}$ account for the *m*-th row and the *n*-th row of $A_{t}$ and $A_{r}$, respectively. Therefore, for the arbitrary array geometry, the following rotationally invariant characteristics exist:
(25a)$$J_{t,m}A\Delta_{t,m}=J_{t,m+1}A$$
(25b)$$J_{r,n}A\Delta_{r,n}=J_{r,n+1}A$$
where
(26a)$$\Delta_{t,m}=\mathrm{diag}\left\{\frac{e^{j2\pi d_{t,m+1,1}/\lambda }}{e^{j2\pi d_{t,m,1}/\lambda }},\frac{e^{j2\pi d_{t,m+1,2}/\lambda }}{e^{j2\pi d_{t,m,2}/\lambda }},\cdots ,\frac{e^{j2\pi d_{t,m+1,K}/\lambda }}{e^{j2\pi d_{t,m,K}/\lambda }}\right\}$$
(26b)$$\Delta_{r,n}=\mathrm{diag}\left\{\frac{e^{j2\pi d_{r,n+1,1}/\lambda }}{e^{j2\pi d_{r,n,1}/\lambda }},\frac{e^{j2\pi d_{r,n+1,2}/\lambda }}{e^{j2\pi d_{r,n,2}/\lambda }},\cdots ,\frac{e^{j2\pi d_{r,n+1,K}/\lambda }}{e^{j2\pi d_{r,n,K}/\lambda }}\right\}$$

Replacing A with $E_{s}T^{-1}$ yields
(27a)$$J_{t,m}E_{s}T^{-1}\Delta_{t,m}T=J_{t,m+1}E_{s}$$
(27b)$$J_{r,n}E_{s}T^{-1}\Delta_{r,n}T=J_{r,n+1}E_{s}$$

Equivalently,
(28a)$$T^{-1}\Delta_{t,m}T={\left(J_{t,m}E_{s}\right)}^{†}J_{t,m+1}E_{s}$$
(28b)$$\Delta_{r,n}=T{\left(J_{r,n}E_{s}\right)}^{†}J_{r,n+1}E_{s}T^{-1}$$

The left part of ([Disp-formula FD28a-sensors-24-06065]) coincides the eigendecomposition of a matrix. Therefore, from the eigendecomposition of ${\left(J_{t,m}E_{s}\right)}^{†}J_{t,m+1}E_{s}$, one can obtain estimates of $\Delta_{t,m}$ and T, which are denoted by ${\hat{\Delta }}_{t,m}$ and $\hat{T}$, respectively. Therefore, by calculating the right side of ([Disp-formula FD28b-sensors-24-06065]) using $\hat{T}$, one can obtain an estimate of $\Delta_{r,n}$, which is denoted by ${\hat{\Delta }}_{r,n}$.

Let $\delta_{t,m,k}$ and $\delta_{r,n,k}$ be the *k*-th diagonals of $\Delta_{t,m}$ and $\Delta_{r,n}$, respectively. Theoretically,
(29a)$$\delta_{t,m,k}=e^{j2\pi \left(d_{t,m+1,k}-d_{t,m,k}\right)/\lambda }$$
(29b)$$\delta_{r,n,k}=e^{j2\pi \left(d_{r,n+1,k}-d_{r,n,k}\right)/\lambda }$$

Accordingly,
(30a)$$\begin{array}{ccc} \mathrm{phase}\left\{\delta_{t,m,k}\right\} & = & 2\pi /\lambda \left(d_{t,m+1,k}-d_{t,m,k}\right)\\ & = & 2\pi /\lambda \left(p_{t,m+1}^{T}-p_{t,m}^{T}\right)c_{t,k} \end{array}$$
(30b)$$\begin{array}{ccc} \mathrm{phase}\left\{\delta_{r,n,k}\right\} & = & 2\pi /\lambda \left(d_{r,n+1,k}-d_{r,n,k}\right)\\ & = & 2\pi /\lambda \left(p_{r,n+1}^{T}-p_{r,n}^{T}\right)c_{r,k} \end{array}$$

In matrix format, we have
(31a)$$\begin{array}{ccc} v_{t,k} & = & \mathrm{phase}\left\{{\left[\delta_{t,1,k},\delta_{t,2,k},\cdots ,\delta_{t,M-1,k}\right]}^{T}\right\}\\ & = & 2\pi /\lambda D_{t}c_{t,k} \end{array}$$
(31b)$$\begin{array}{ccc} v_{r,k} & = & \mathrm{phase}\left\{{\left[\delta_{r,1,k},\delta_{r,2,k},\cdots ,\delta_{r,N-1,k}\right]}^{T}\right\}\\ & = & 2\pi /\lambda D_{r}c_{r,k} \end{array}$$
where
(32a)$$D_{t}=\left[\begin{array}{c} p_{t,2}^{T}-p_{t,1}^{T}\\ p_{t,3}^{T}-p_{t,2}^{T}\\ \vdots \\ p_{t,M}^{T}-p_{t,M-1}^{T} \end{array}\right]$$
(32b)$$D_{r}=\left[\begin{array}{c} p_{r,2}^{T}-p_{r,1}^{T}\\ p_{r,3}^{T}-p_{r,2}^{T}\\ \vdots \\ p_{r,N}^{T}-p_{r,N-1}^{T} \end{array}\right]$$

As a result, $c_{t,k}$ and $c_{r,k}$ can be estimated via
(33a)$${\hat{c}}_{t,k}=\lambda /2\pi D_{t}^{†}{\hat{v}}_{t,k}$$
(33b)$${\hat{c}}_{r,k}=\lambda /2\pi D_{r}^{†}{\hat{v}}_{r,k}$$
with
(34a)$${\hat{v}}_{t,k}=\mathrm{phase}\left\{{\left[{\hat{\delta }}_{t,1,k},{\hat{\delta }}_{t,2,k},\cdots ,{\hat{\delta }}_{t,M-1,k}\right]}^{T}\right\}$$
(34b)$${\hat{v}}_{r,k}=\mathrm{phase}\left\{{\left[{\hat{\delta }}_{r,1,k},{\hat{\delta }}_{r,2,k},\cdots ,{\hat{\delta }}_{r,N-1,k}\right]}^{T}\right\}$$
where ${\hat{\delta }}_{t,m,k}$ and ${\hat{\delta }}_{r,n,k}$ denote the *k*-th entities of ${\hat{\Delta }}_{t,m}$ and ${\hat{\Delta }}_{r,n}$, respectively. Unfortunately, when the inter-element space of two adjacent sensors is larger than λ/2, the values of ${\hat{v}}_{t,k}$ and ${\hat{v}}_{r,k}$ obtained from (34) may not correctly reveal the true phases due to the periodic ambiguity of the circular function. To accurately recover the phases, there is a need to compensate for the error before further calculation. In fact, there exists
(35a)$$v_{t,k}=2\pi /\lambda D_{t}c_{t,k}+2\pi {ȷ}_{t,k}$$
(35b)$$v_{r,k}=2\pi /\lambda D_{r}c_{r,k}+2\pi {ȷ}_{r,k}$$
where ${ȷ}_{t,k}$ and ${ȷ}_{r,k}$ are real-valued integer vectors. In this paper, we suppose that there are $M_{1}$ pairs of Tx sensors and $N_{1}$ pairs of Rx sensors on a straight line with inter-element spacing of less than or equal to λ/2; then, we compute the standard phase factors as
(36a)$${\tilde{c}}_{t,k}=\lambda /2\pi {\tilde{D}}_{t}^{†}{\tilde{v}}_{t,k}$$
(36b)$${\tilde{c}}_{r,k}=\lambda /2\pi {\tilde{D}}_{r}^{†}{\tilde{v}}_{r,k}$$
where ${\tilde{D}}_{t}$, ${\tilde{D}}_{r}$, ${\tilde{v}}_{t,k}$, and ${\tilde{v}}_{r,k}$ denote the matrices and vectors corresponding to the previously mentioned sensors. Therefore, we can calculate the estimated phase differences via
(37a)$${\bar{v}}_{t,k}=2\pi /\lambda D_{t}{\tilde{c}}_{t,k}$$
(37b)$${\bar{v}}_{r,k}=2\pi /\lambda D_{r}{\tilde{c}}_{r,k}$$

Moreover, integer vectors can be estimated via
(38a)$${\hat{ȷ}}_{t,k}=\mathrm{round}\left\{\frac{{\hat{v}}_{t,k}-{\bar{v}}_{t,k}}{2\pi }\right\}$$
(38b)$${\hat{ȷ}}_{r,k}=\mathrm{round}\left\{\frac{{\hat{v}}_{r,k}-{\bar{v}}_{r,k}}{2\pi }\right\}$$

Phase compensations can be accomplished via
(39a)$${\overset{˘}{v}}_{t,k}={\hat{v}}_{t,k}+2\pi {\hat{ȷ}}_{t,k}$$
(39b)$${\overset{˘}{v}}_{r,k}={\hat{v}}_{r,k}+2\pi {\hat{ȷ}}_{r,k}$$

Then, ${\hat{v}}_{t,k}$ and ${\hat{v}}_{r,k}$ are replaced with ${\overset{˘}{v}}_{t,k}$ and ${\overset{˘}{v}}_{r,k}$, respectively, the right side of (33) is computed. Finally, the 2D DOD and 2D DOA can be estimated via
(40a)$${\hat{\phi }}_{t,k}=\arctan\left\{{\hat{c}}_{t,k}\left(2\right)/{\hat{c}}_{t,k}\left(1\right)\right\}$$
(40b)$${\hat{\theta }}_{t,k}=\arccos\left\{{\hat{c}}_{t,k}\left(3\right)\right\}$$
(40c)$${\hat{\phi }}_{r,k}=\arctan\left\{{\hat{c}}_{r,k}\left(2\right)/{\hat{r}}_{t,k}\left(1\right)\right\}$$
(40d)$${\hat{\theta }}_{r,k}=\arccos\left\{{\hat{c}}_{r,k}\left(3\right)\right\}$$
where ${\hat{c}}_{t,k}\left(q\right)$ and ${\hat{c}}_{r,k}\left(q\right)$ represent the *q*-th entities of ${\hat{c}}_{t,k}$ and ${\hat{c}}_{r,k}$, respectively.

Now, we have achieved the objective of the proposed normalized ESPRIT algorithm, which can be briefly summarized by the following steps:

**Step 1**. Calculate the covariance matrix (${\hat{R}}_{y}$) via (Equation 12);**Step 2**. Perform eigenvalue decomposition on ${\hat{R}}_{y}$ to obtain the signal subspace (${\hat{E}}_{s}$);**Step 3**. Construct the selection matrices ($J_{t,m}$ and $J_{r,n}$) according to (24), and perform eigenvalue decomposition on ${\left(J_{t,m}E_{s}\right)}^{†}J_{t,m+1}E_{s}$ to obtain ${\hat{\Delta }}_{t,m}$ and $\hat{T}$;**Step 4**. Calculate the right side of ([Disp-formula FD28b-sensors-24-06065]) using $\hat{T}$ to obtain ${\hat{\Delta }}_{r,n}$;**Step 5**. Obtain ${\bar{v}}_{t,k}$ and ${\bar{v}}_{r,k}$ via (37), and compensate for the phase using (39);**Step 6**. Compute the right side of (33) and obtain the 2D DOD and 2D DOA via (40).

## 4. Algorithmic Analyses

### 4.1. Identifiability

Identifiability refers to the maximum number of targets that the algorithm can identify. The traditional RD-MUSIC algorithm can identify a maximum of MN targets. The identifiability of the proposed algorithm is constrained by the maximum ranks of $\Delta_{t,m}$ and $\Delta_{r,n}$. According to (27), the ranks of the two matrices are smaller than *M* and *N*; thus, the maximum identifiable target number (*K*) must be smaller than $\min\left\{M,N\right\}$. Obviously, the RD-MUSIC algorithm exhibits better identifiability than the proposed algorithm.

### 4.2. Complexity

Complexity refers to the number of complex multiplications required in each algorithm. The calculation of the covariance matrix requires $O\left\{M^{2}N^{2}L\right\}$ complex multiplications, and the eigendecomposition of the covariance matrix requires $O\left\{M^{3}N^{3}\right\}$ complex multiplications. Peak searching in the traditional RD-MUSIC algorithm, requires $O\left\{M^{2}N^{2}Q\right\}$ complex multiplications, where *Q* is for the grid number. The main calculation of the proposed algorithm is the estimation of $\Delta_{t,m}$ and $\Delta_{r,n}$, which requires a total of $O\left\{MK^{2}+NK^{2}+K^{3}\right\}$ complex multiplications. Since *Q* is usually much larger than MN, RD-MUSIC always suffers a considerably greater computational burden than the proposed algorithm.

### 4.3. Cramer–Rao Bounds (CRBs)

The deterministic CRBs on joint 2D DOA and 2D DOA estimation are expressed as
(41)$$CRB=\frac{\sigma^{2}}{2L}{\left[\mathrm{real}\left\{\left({\tilde{A}}^{H}\Pi_{A}^{⊥}\tilde{A}\right)⊕\left(R_{s}^{T}⊗1_{4\times 4}\right)\right\}\right]}^{-1},$$
where $\tilde{A}=\left[A_{\theta_{t}},A_{\phi_{t}},A_{\theta_{r}},A_{\phi_{r}}\right]$, $A_{\theta_{t}}=\left[\frac{\partial a_{1}}{\partial \theta_{t,1}},\frac{\partial a_{2}}{\partial \theta_{t,2}},\cdots ,\frac{\partial a_{K}}{\partial \theta_{t,K}}\right]$, $A_{\phi_{t}}=\left[\frac{\partial a_{1}}{\partial \phi_{t,1}},\frac{\partial a_{2}}{\partial \phi_{t,2}},\cdots \right.$, $\left.\frac{\partial a_{K}}{\partial \phi_{t,K}}\right]$, $A_{\theta_{r}}=\left[\frac{\partial a_{1}}{\partial \theta_{r,1}},\frac{\partial a_{2}}{\partial \theta_{r,2}},\cdots ,\frac{\partial a_{K}}{\partial \theta_{r,K}}\right]$, $A_{\phi_{r}}=\left[\frac{\partial a_{1}}{\partial \phi_{r,1}},\frac{\partial a_{2}}{\partial \phi_{r,2}},\cdots ,\frac{\partial a_{K}}{\partial \phi_{r,K}}\right]$, $\Pi_{A}^{⊥}=I-{\mathrm{AA}}^{†}$, and $1_{4\times 4}$ represents a 4×4 full ones matrix.

## 5. Simulation Results and Discussion

The efficacy of the proposed algorithmic approach is validated through extensive Monte Carlo simulations. The simulation setup considers a bistatic MIMO system with an arbitrary geometry, featuring a Tx array with *M* elements, an Rx array with *N* elements, and K=3 far-field targets located within the same range. The target reflection characteristics are modeled according to the Swerling II model ($s\left(\tau \right)$ satisfies a complex, random Gaussian process with means zeroed and the covariance matrix scaled to the identical matrix), and for estimation purposes, *L* snapshots are acquired. To evaluate the algorithm’s accuracy and precision in estimating the target parameters, we adopt the root mean square error (RMSE) metric.

In the first simulation, we illustrate the scatting results of the proposed algorithm, where M=6, N=8, L=200, and the signal-to-noise ratio (SNR) is set to 20 dB. In this simulation, we consider a 3D array geometry for both the Tx array and the Rx array. The entities of the Tx position vector and the Rx position vector are randomly distributed within ranges of [0,3λ] and [0,4λ], respectively. The 2D DOD and 2D DOA are fixed at $\theta_{t}=\left(20^{\circ },50^{\circ },40^{\circ }\right)$, $\phi_{t}=\left(25^{\circ },35^{\circ },45^{\circ }\right)$, $\theta_{r}=\left(10^{\circ },25^{\circ },45^{\circ }\right)$, and $\phi_{r}=\left(15^{\circ },35^{\circ },55^{\circ }\right)$, respectively. A graphical depiction, obtained through the application of our innovative algorithm is presented in Figure 2. Evidently, the angle scatter points of the proposed algorithm are tightly clustered around the correct points. This indicates that the proposed method achieves automatic pairing and high-precision 2D DOD and DOA estimation for conformal-array MIMO radar.

In the second simulation, we compare the performances of various algorithms at different SNRs, where M=6, N=8, and L=200. Herein, we consider a 1D linear array at both the Tx end and the Rx end. The positions of the Tx sensors and the Rx sensors are randomly chosen from within the ranges of [0,5M] (λ/2) and [0,5N] (λ/2), respectively. The DOD and DOA are fixed at $\theta_{t}=\left(20^{\circ },30^{\circ },40^{\circ }\right)$, and $\phi_{t}=\left(25^{\circ },35^{\circ },45^{\circ }\right)$, respectively. For comparison purposes, we evaluate the performances of RD-MUISC, ESPRIT, and CRB, with the search grid of RD-MUSIC set to [0,90] with an interval of $0.01^{\circ }$. Figure 3 illustrates the results. Clearly, the RMSE values for DOD estimation achieved by RD-MUSIC consistently surpass those of the proposed method over the full range of SNRs, but RD-MUSIC fails to work in DOA estimation. The main reason that RD-MUSIC outperforms the proposed algorithm in DOD estimation is that in RD-MUSIC, spectrum searching is insensitive to the array geometry, and it can utilize the full virtual aperture (MN, units) of the MIMO radar. However, the effective aperture of the proposed algorithm is only *N* (units). Since the LS in RD-MUSIC is only suitable for linear-array geometry, it fails to work in DOA estimation. Moreover, the traditional ESPRIT algorithm is only suitable for uniform Tx/Rx arrays, and the inter-element spacing must not be greater than a half wavelength; thus, it always fails in such cases. Thanks to phase compensation in the proposed algorithm, it can correctly operate under such an array configuration.

In the third simulation, we repeat the comparison with different snapshot values (*L*) and the SNR set to 10dB relative to other setups that are the same as in the previous simulation. The results are exhibited in Figure 4. As theoretically expected, the RMSE changes in such a manner that the estimation performance improves with the increase in effective data. Notably, the proposed method consistently outperforms the compared algorithms in DOA estimation, but it provides worse performance than RD-MUSIC in DOD estimation.

In the forth simulation, we repeat the comparison with different inter-element spacing (*d*). Similarly, other parameters are fixed as in the second simulation. The results are displayed in Figure 5. An interesting observed phenomenon is that the ESPRIT algorithm offers the same RMSE performance as the proposed algorithm for the ULA configuration (d=0.5λ), while RD-MUSIC offers more accurate results than the proposed algorithm in the presence of the ULA configuration. However, when *d* is larger than 0.5λ, both RD-MUSIC and ESPRIT fail. As expected, the RMSE of the proposed algorithm is improved with the increasing *d*, as a larger *d* value is associated with a larger array aperture, yielding more accurate results.

Finally, we repeat the comparison with different numbers of Rx sensors (*N*). Similarly, other parameters are fixed as in the second simulation. The RMSE results are displayed in Figure 6. We can observe that the RMSE of the proposed algorithm decreases with increasing *N*. Similarly, RD-MUSIC is only effective in DOD estimation, and ESPRIT fails in both DOD and DOA estimation. Moreover, we compare the average running time with a simulation was conducted on a an HP Z840 workstation featuring dual Intel?Xeon? E5-2650 v4 processors with 2.20 GHz and 128 GB DDR4 RAM, respectively, utilizing MATLAB R2016a. Figure 7 presents the results. As expected, both ESPRIT and the proposed algorithm are less sensitive to *N*, while RD-MUSIC requires more time for the calculation. Moreover, the running time of RD-MUSIC is much longer than that of the proposed algorithm, proving that the proposed algorithm is more efficient than RD-MUSIC.

## 6. Conclusions

In this paper, we introduce an estimator based on ESPRIT for 2D DOD and DOA estimation in a conformal-array MIMO radar system. The cornerstone of the estimator lies is the rotational invariance of the virtual response matrix. Following the eigenvalue decomposition of the covariance matrix, we obtain the normalized rotational factor matrices corresponding to the Tx array and the Rx array. Subsequently, 2D DOD and DOA estimation is accomplished by utilizing least squares phase compensation and the LS fitting technique. Simulation results show that the proposed algorithm can provide accurate estimation results for conformal-array MIMO radar, even if several sensors have inter-element spacing larger than a half wavelength. Moreover, the results show that the running time of the proposed algorithm is only 1%–10% that of the RD-MUSIC algorithm. The flexibility of the array geometry and the closed-form results make these estimators promising candidates for future applications in various domains, including wireless communication and navigation, among others. However, there is still room for further improvements to handle more complex scenarios, such as mutual sensor coupling, gain-phase errors, etc.

## Figures and Tables

**Figure 1 sensors-24-06065-f001:**
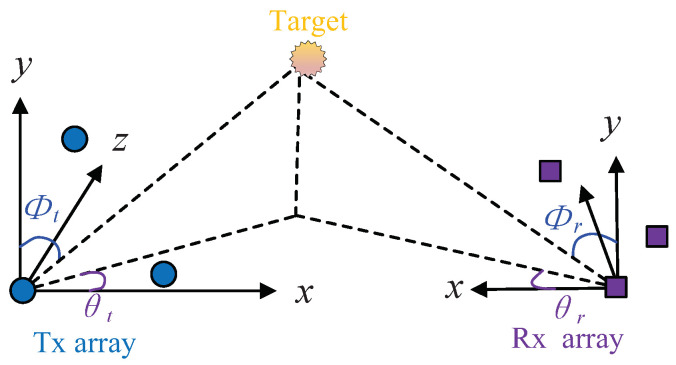
Bistatic conformal-array MIMO radar scenario.

**Figure 2 sensors-24-06065-f002:**
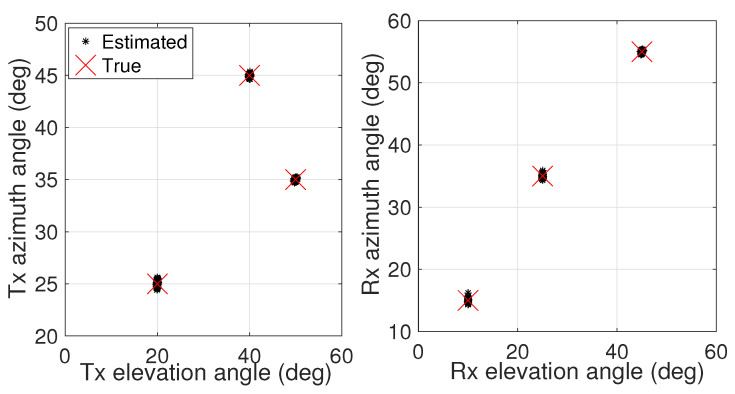
Scatting results of the true 2D angles and the estimated 2D angles obtained via the proposed algorithm. (**Left**) 2D DOD results; (**right**) 2D DOA results.

**Figure 3 sensors-24-06065-f003:**
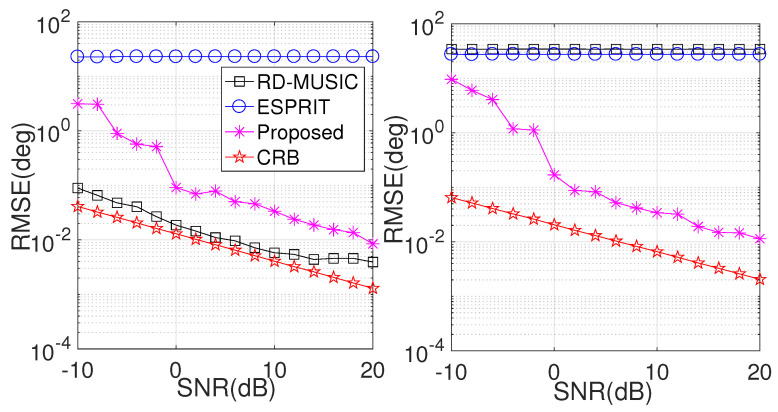
Performance comparison versus SNR. (**Left**) RMSE comparison in DOD estimation (**right**) RMSE comparison in DOA estimation.

**Figure 4 sensors-24-06065-f004:**
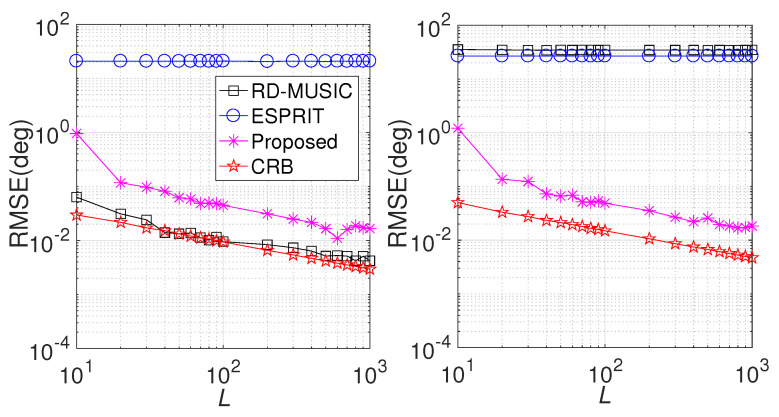
Performance comparison versus *L*. (**Left**) RMSE comparison in DOD estimation; (**right**) RMSE comparison in DOA estimation.

**Figure 5 sensors-24-06065-f005:**
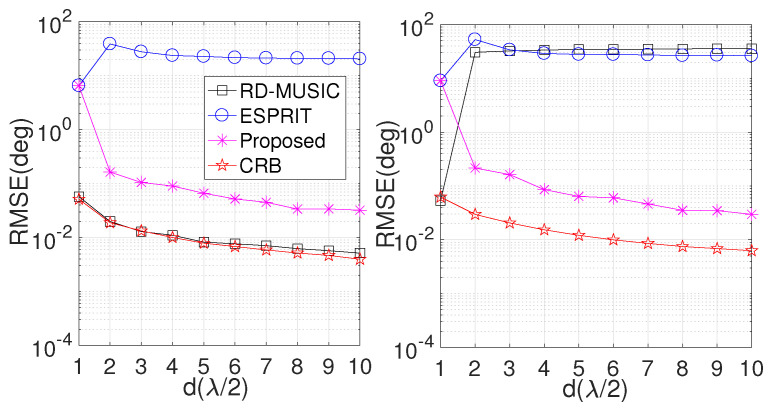
Performance comparison versus inter-element spacing (*d*). (**Left**) RMSE comparison in DOD estimation; (**right**) RMSE comparison in DOA estimation.

**Figure 6 sensors-24-06065-f006:**
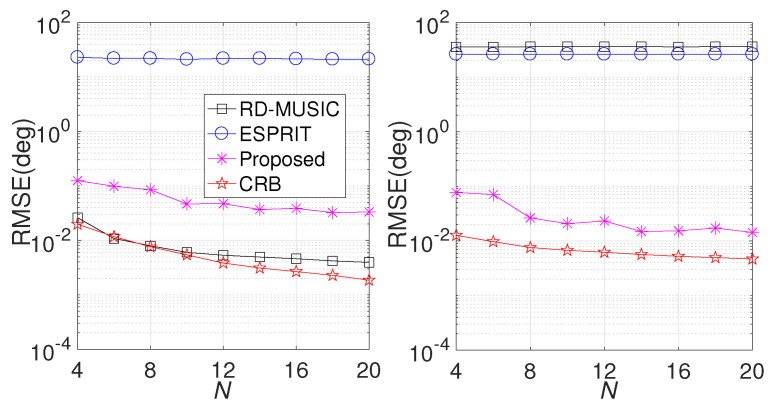
Performance comparison versus *N*. (**Left**) RMSE comparison in DOD estimation; (**right**) RMSE comparison in DOA estimation.

**Figure 7 sensors-24-06065-f007:**
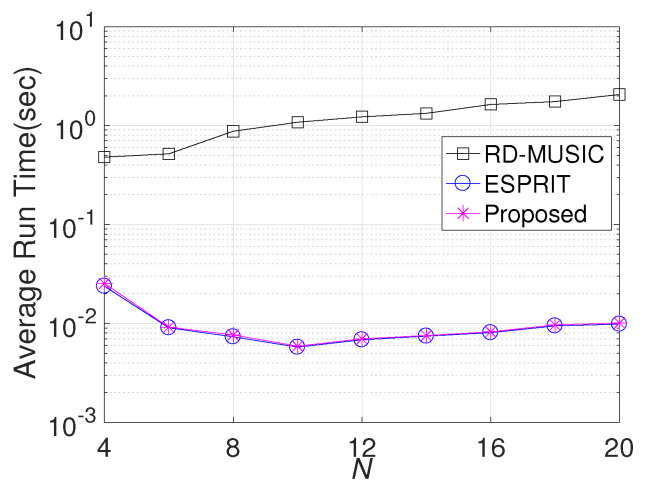
Average running time comparison versus *N*. Obviously, the proposed algorithm requires only 1–10% of the running time of RD-MUSIC.

## Data Availability

The original contributions presented in the study are included in the article, further inquiries can be directed to the corresponding author.

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
