# Peer review of "Computationally Efficient Direction Finding for Conformal MIMO Radar"

_sensors, 2024, doi:10.3390/s24186065_

Round 1
Reviewer 1 Report
Comments and Suggestions for Authors
This paper deals with direction finding in MIMO radar, and proposed an ESPRIT estimator that is suitable for an arbitrary array geometry. This paper shows algorithmic novelty, I recommend it to be accepted subject to minor revision.
Comment 1. One key consideration is the determination of the source number, K, which appears to be treated as a known prior in the descriptions.
Comment 2. It is crucial to clarify whether the proposed estimator is suitable for coherent sources, emphasizing this aspect within the context of the discussion.
Comment 3. As demonstrated, the matrix T can be estimated, yet its role in parameter estimation remains unexplored. It is essential to elucidate the physical significance of T and its impact on parameter estimation.
Comment 4. In the simulation section, it is imperative to provide a comprehensive explanation of the simulated sources within the context of the study, enhancing the clarity of the experimental setup.
Reviewer 2 Report
Comments and Suggestions for Authors
See attachment

Require minor editing.
Reviewer 3 Report
Comments and Suggestions for Authors
In this work, an improved rotational invariant technique is proposed. The core of the proposed algorithm is to estimate the phase differences between the adjacent sensors, and then it eliminates the phase ambiguity via the previous estimated standard phase difference. Thereafter, DODs and DOAs are obtained via Least Squares (LS) fitting. The following issues are suggested to address.
1. The authors proposed to avoid the phase ambiguity caused by large inter-element spacing using ESPRIT. However, in conformal array MIMO radar systems, the rotational invariant technique is generally not satisfied. How to address this issue?
2. The authors may include a discussion comparing their work with recent MIMO radar studies [R1] that also employ DOA and DOD estimation. This comparison would help to better highlight the strengths of the current study.
[R1] Transmit Array Interpolation for DOA Estimation via Tensor Decomposition in 2-D MIMO Radar
3. In simulations, the performance of proposed method is lower than that of RD-MUSIC in DOD estimation, but better than that for DOA estimation. Please clarify this issue.
4. The recent work also employs the idea of rotational invariant for angle parameters estimation in [R2]. The authors may discuss the advantage of the proposed method over this work.
[R2] Channel Estimation for Movable-Antenna MIMO Systems Via Tensor Decomposition. 10.1109/LWC.2024.3450592]
5. Minor issues:
In abstract, “a improved rotational invariant technique”;
Page 2, line 60, “inter-emement spacing”
Line 224, “The deterministic CRBs on joint 2D-DOA and 2D-DOA”
Comments on the Quality of English Language
see comments
Reviewer 4 Report
Comments and Suggestions for Authors
This study presents an improved ESPRIT-based technique for efficient direction finding in conformal MIMO radar systems. The proposed method addresses computational inefficiencies and phase ambiguity in joint DOD and DOA estimation by estimating and compensating phase differences between adjacent sensors. Validated through Monte Carlo simulations, the algorithm achieves high accuracy and efficiency, outperforming traditional methods, especially in complex, non-uniform array geometries.
Recommendation: The manuscript is promising but requires revisions before it can be accepted for publication. I suggest a Major Revision to address the points mentioned below.
1. Abstract
The abstract is concise and provides a good overview; however, it could include more specific details about the simulation results to give the reader a clearer sense of the study's outcomes.
2. Introduction
- Pages 2-3: The introduction is well-structured and provides a good context for the study. However, it could benefit from a clearer articulation of the research gap. The authors discuss existing methods but should more explicitly emphasize how their proposed method advances the field.
- Pages 3-4: The literature review is comprehensive and includes relevant citations. However, it could be improved by including a critical analysis of the cited works. For instance, the discussion could focus more on the limitations of the current methods and how the proposed approach addresses these issues. Additionally, some key references are missing that could strengthen the context of the study, particularly recent advancements in MIMO radar signal processing.
3. Methodology
The methodology is robust and detailed, which is a strong point of the paper. The authors provide a clear explanation of the proposed ESPRIT algorithm and its enhancements. However, there are a few areas where additional clarification would be helpful. For example, the authors should provide more details on how the phase compensation strategy is implemented in practice. Moreover, the mathematical notations, while generally clear, could benefit from a brief glossary or table summarizing key symbols for ease of reference, rather than the paragraph provided at the end of the introduction.
4. Experimental Setup and Results
The use of Monte Carlo simulations to validate the algorithm is appropriate and thorough. However, the authors should include a more in-depth discussion of the limitations of their approach, particularly regarding the assumptions made in the simulations. For example, how would the algorithm perform in more complex scenarios with non-ideal conditions? Additionally, the comparison with other algorithms is a strong point, but the authors should discuss why certain algorithms (e.g., RD-MUSIC) underperform in specific cases.
5. Figures
The figures and tables are generally well-organized and effectively support the text. However, there are a few issues that need to be addressed:
- Fig. 2 (Page 9): The captions should be more descriptive. For example, the authors should explain what the "scatting results" refer to and how they validate the proposed algorithm.
- Fig. 5 (Page 11): The captions are hiding some points of the ESPRIT plot. Please consider changing its position.
- Fig. 7 (Page 11): The authors should add a brief discussion of what the running time comparison implies about the efficiency of their algorithm relative to RD-MUSIC.
6. Conclusion
The conclusion must summarize the main contributions of the paper. Please consider adding the main quantitative findings. Moreover, it could be strengthened by explicitly stating the practical implications of the proposed method. For instance, how might this approach be applied in real-world radar systems? The authors should also suggest specific directions for future research, such as extending the method to handle more complex scenarios or different types of array geometries.
Comments on the Quality of English LanguageGrammatical Errors
To improve the grammatical accuracy of the paper, a thorough proofreading is recommended. Here are some examples of errors:
- On Page 1: "In this paper, a improved rotational invariant technique is proposed."
Correction: "In this paper, an improved rotational invariant technique is proposed."
- On Page 2: "A MIMO radar transmit mutual orthogonal waveforms via multiple active antennas..."
Correction: "A MIMO radar transmits mutually orthogonal waveforms via multiple active antennas..."
- On Page 6: "This paper re-visit the problem of joint DOD and DOA estimation..."
Correction: "This paper revisits the problem of joint DOD and DOA estimation..."
- On Page 11: "Moreover, the runing time of RD-MUSIC is much larger than the proposed algorithm..."
Correction: "Moreover, the running time of RD-MUSIC is much larger than the proposed algorithm..."
- On Page 11: "An interesting phenomenon observed is that the ESPRIT algorithm offers the same RMSE performance to the proposed algorithm..."
Correction: "An interesting phenomenon observed is that the ESPRIT algorithm offers the same RMSE performance as the proposed algorithm..."
Reviewer 5 Report
Comments and Suggestions for Authors
In this paper, the authors have developed an estimator for joint DOD and DOA estimation in bistatic MIMO radar. The core of the proposed algorithm is to estimate the normalized rotational invariance factors between the Tx array and the Rx array. Since it rely on the inter-element position information rather than the uniformity of the array, the proposed algorithm is suitable for an arbitrary array geometry. From my perspective, this article shows novelty in algorithm design, it can be accepted after necessary minor revisions.
Here are my concerns:
(1). In the introduction section, both of the advantages and disadvantages of ESPRIT should be stressed. So that the reader can understand why such approach is so important for a radar system.
(2). The format of Title 3 is incorrect, and the same issue exists with other titles. Please check.
(3). The general array manifold exponential form has a negative sign in the exponent, \( e^{-j2\pi**}. Is there a specific reason the author removed the negative sign?
(4). The reference format might be incorrect. Please revise according to the latest MDPI template.
Round 2
Reviewer 3 Report
Comments and Suggestions for Authors
No comments
Reviewer 4 Report
Comments and Suggestions for Authors
The authors have addressed all comments.